# Physiological Alterations and Nondestructive Test Methods of Crop Seed Vigor: A Comprehensive Review

Muye Xing [1,2,3,4,5], Yuan Long [2,3,4,5], Qingyan Wang [2,3,4,5], Xi Tian [2,3,4,5], Shuxiang Fan [2,3,4,5], Chi Zhang [2,3,4,5] and Wenqian Huang [2,3,4,5,*]

1    School of Agricultural Engineering, Jiangsu University, Zhenjiang 212013, China
2    Intelligent Equipment Research Center, Beijing Academy of Agriculture and Forestry Sciences, Beijing 100097, China
3    National Research Center of Intelligent Equipment for Agriculture, Beijing 100097, China
4    Key Laboratory of Agri-Informatics, Ministry of Agriculture, Beijing 100097, China
5    Beijing Key Laboratory of Intelligent Equipment Technology for Agriculture, Beijing 100097, China
*    Correspondence: huangwq@nercita.org.cn

**Abstract:** Seed vigor is one of the essential contents of agricultural research. The decline of seed vigor is described as an inevitable process. Recent studies have shown that the oxidative damage caused by reactive oxygen species (ROS) is the main reason for the destruction of various chemicals in seeds and eventually evolves into seed death. The traditional vigor tests, such as the seed germination test and TTC staining, are commonly used to assess seed vigor. However, these methods often need a large number of experimental samples, which will bring a waste of seed resources. At present, many new methods that are fast and nondestructive to seeds, such as vibrational spectroscopic techniques, have been used to test seed vigor and have achieved convincing results. This paper is aimed at analyzing the microchanges of seed-vigor decline, summarizing the performance of current seed-vigor test methods, and hoping to provide a new idea for the nondestructive testing of a single seed vigor by combining the physiological alterations of seeds with chemometrics algorithms.

**Keywords:** physiological factors; novel test; mechanism

## 1. Introduction

Seeds are the most basic production resources in agriculture [1], the most momentous and fundamental part of the life cycle of various crops, and the core of various technical measures in agricultural science. In the face of population growth and the repeated outbreak of COVID-19, the output of grain and other crops needs to be stable. The quality of seeds will have a significant impact on the yield of crops, which is often described by the potential of germination and seed vigor. Generally, the decline of seed vigor is caused by its internal regulation or the influence of the external environment [2].

Even if seeds are stored in the most appropriate state, the vigor of the seeds will owly decline [3]. This process is often called seed aging. Existing studies have shown that reactive oxygen species (ROS) may be the culprit leading to a series of physiological alterations, reducing seed vigor and eventually causing seed death [4]. It is reported that a large number of outbreaks of ROS will attack lipids, proteins, and genes in cells and make their functions abnormal [5]. The damaged lipid in the membrane will produce lipid peroxide molecules and lead to the change of membrane permeability, which is a key event in the process of seed-vigor decline [6]. During this period, the nutrients in the membrane conducive to plant growth may penetrate into the matrix and affect the normal physiological activities of seeds [5]. Proteins play an irreplaceable role in cell function. The modification of proteins by ROS is also considered to be a representative event of oxidative damage to seeds, and this may be a type of permanent damage that cannot be repaired [6]. The direct consequence is that a large number of modified proteins accumulate and affect

the normal function of cells [7]. Genes are described as the core of seeds and are responsible for regulating their own physiological conditions according to external conditions. ROS accumulation will cause abnormal gene expression and eventually cause seed death [8]. However, the above changes seem to be unclear and confusing for researchers who are just starting to work on seed-vigor analysis. To solve this situation, this paper will briefly review the research status of physiological changes in the process of seed aging. The purpose is to explain the continuous physiological process of seed-vigor deterioration as much as possible, to discuss the causes of the loss of the ability for seeds to germinate, and to provide a biological basis for and to understand the mechanism of the nondestructive test of seed vigor.

If low-vigor seeds are used in agricultural production, abnormal seedlings or nongerminating seeds may be harvested, which will waste the manpower and financial resources of seed cultivation [3]. Reasonable and effective seed-vigor screening technology can test and eliminate low-vigor or dead seeds, so as to improve agricultural yield. The traditional seed-vigor test has the characteristics of being a simple test procedure and has stable results, high accuracy, and strong intuition, but it needs more experimental samples and will cause damage and waste. In recent years, spectral technology has attracted the attention of many scholars [9–16]. It has the unique advantages of being strongly objective, having excellent reproducibility, requiring generally no pretreatment, and being an onsite testing procedure, which makes it an indispensable technical method for product quality testing for such procedures as the seed-vigor test, seed mildew test, and so on. In particular, combined with hyperspectral imaging technology (HSI), the method of spectral integration analyzes the test data more comprehensively and brings better test results [17,18]. In addition, other methods for testing seed vigor have emerged one after another [19,20]. By and large, compared with the traditional 2,3,5-triphenyltetrazolium chloride (TTC) dyeing and germination methods, the combination of new scientific and technological means and new classification algorithms has the characteristics of being fast, simple, causing no damage to the seed structure, and achieving a quite high accuracy. In order to provide assistance to future researchers in the domain of seed-viability testing, this paper will review several seed-vigor test methods in recent years for understanding the principle of existing seed-vigor testing and exploring future trends.

Specifically, our review is divided into the following chapters. (1) We discuss physiological changes in the decline of seed vigor caused by ROS and how these changes affect seed vigor in turn. (2) We introduce the current vigor test methods for seeds in the process of aging or vigor decline. (3) We make some suggestions for the future research on seed vigor testing.

## 2. Physiological Factors Affecting Seed Vigor

The aging of seeds is an important reason for the decline of seed vigor, and this process is described as an irreversible common phenomenon that occurs after the seeds reach their highest point of maturity physiologically [21]. The decline of seed vigor includes the process of programmed cell death [22], reduction of ROS [4], lipid peroxidation [23], protein carbonylation [24], and the loss of enzyme activity [25]. The main process of ROS in the decline of seed vigor is shown in Figure 1. In seed science, seeds are usually divided into orthodox seeds and stubborn seeds according to their resistance to drying during storage [26]. Most commonly, grain seeds—for example maize (*Zea mays* L.), rice (*Oryza sativa* L.), soybeans (*Glycine max* (L.) *Merr*), barley (*Hordeum vulgare* L.), and wheat (*Triticum aestivum* L.)—all belong to the category of orthodox seeds [27]. Many authors believe that the physiological alterations of orthodox seeds in the process of seed aging may be the same. The ultimate impact of these physiological alterations on seeds is a decline in the vigor of seeds [27].

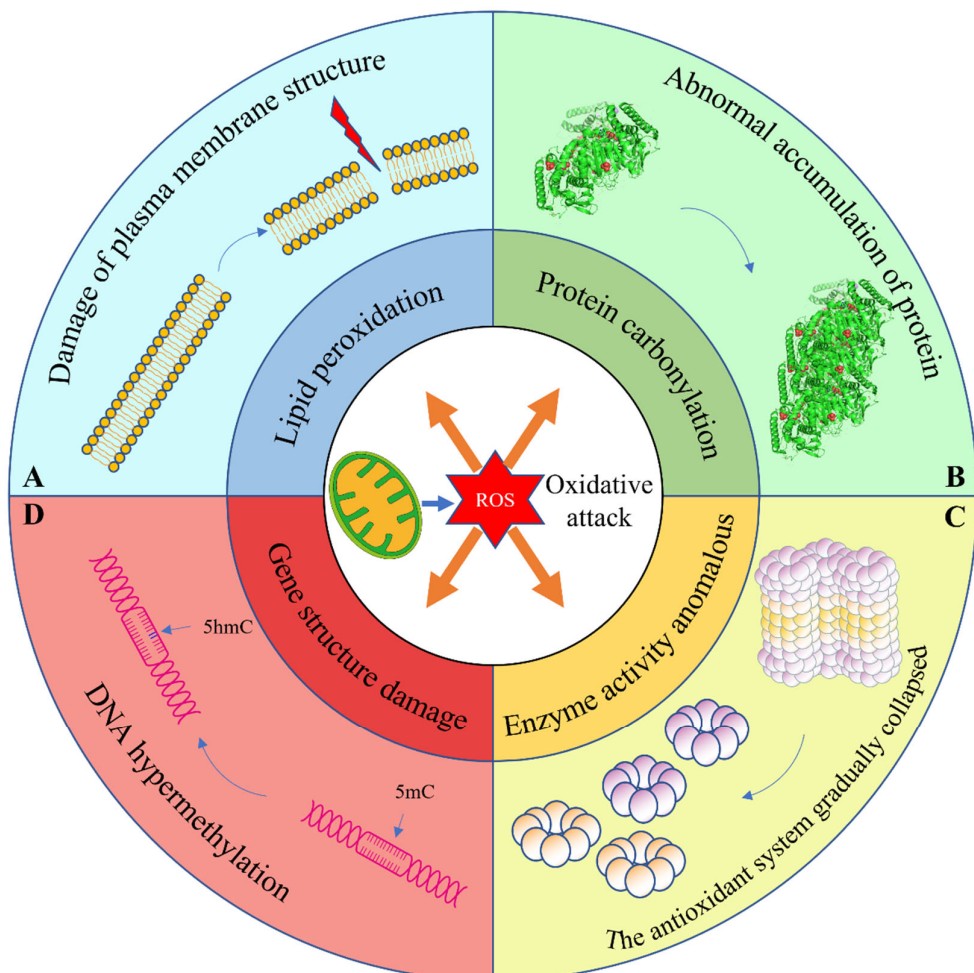

**Figure 1.** Role of ROS in the decline of seed vigor. (**A**) Lipids in plasma membranes are oxidized and decomposed by ROS attack. (**B**) Malondialdehyde (MDA) and 4-hydroxy-non-2-enal (HNE) produced by lipid peroxidation oxidize protein and damage protein function. The protein data bank (PDB) diagram of protein structure is quoted from Huang, et al. [28] PDB ID: 5NUF. (**C**) Abnormal enzyme activity and breakdown of the antioxidant enzyme system. (**D**) The gene methylation balance was broken and hypermethylation occurred, in which 5mC was methylated to 5hmC.

## 2.1. Production of Reactive Oxygen Species (ROS)

During the storage of seeds, the generation of internal ROS is mainly related to the storage conditions of seeds. ROS can be produced everywhere in seed cells, but mainly from the mitochondria in cells [29]. According to the research of Ratajczak, et al. [6], it was found that more ROS were produced in the embryo, especially in the embryonic axis of the seed. This may be closely related to the decline of seed vigor. Subsequent experiments on the protection of embryos with ROS antioxidants also confirmed this view [30]; the principle of the experiment is that the higher the concentration of antioxidants in the embryonic axis, the stronger the activity of enzyme scavengers, so as to reduce the effect of ROS on cell activity.

The production of ROS will have a destructive effect on the lipid, protein, and DNA of cells, which is the key reason for the loss of seed vigor. In addition, the decline of antioxidant enzyme activity also intensifies the decline of seed vigor. Antioxidant enzymes mainly include superoxide dismutase (SOD), catalase (CAT), peroxidase (POD), and glutathione reduce (GR), which constitute the first line of defense against ROS attack [31]. According to Yin et al. [3], the excessive accumulation of ROS will lead to the decrease or loss of ROS scavenging enzyme activity, and the first line of defense against ROS will collapse. Some scholars described the phenomenon when the balance between ROS and ROS elimination

enzymes is broken as oxidative stress [32,33]. The loss of ROS scavenging enzyme activity may even affect the rapid outbreak of ROS in the late stage of seed aging [34]. In this state, ROS will destroy the respiratory system of seeds by attacking mitochondria and cause irreversible damage to seeds.

In addition, a series of alterations, such as lipid peroxidation, protein carbonylation and Maillard reaction during the decline of seed vigor, have been confirmed to be related to the outbreak of ROS.

### 2.2. Lipid Peroxidation

When oxidative stress occurs, the dynamic balance between ROS and ROS elimination enzymes is broken, and ROS begins to erupt. At this time, $H_2O_2$, one of the main components of ROS, begins to attack the plasma membrane structure in the cell and form an oxygen free radical chain reaction, which is called lipid peroxidation [35]. One of the main manifestations of lipid peroxidation is the change of cell membrane permeability, which is also an important event in the process of seed-vigor decline [5,6]. According to the hypothesis made by Agmon, et al. [36] simulating the process of cell ferroptosis, the change of membrane permeability is likely to be the change of membrane characteristics caused by the deterioration after lipid peroxidation. From the perspective of molecular dynamics, the phospholipid part of the cell membrane will change in structure and characteristics after being oxidized, and the generated peroxide will have stronger binding to the hydrogen bond in water. This makes the cell membrane more hydrophilic. Therefore, it is considered that similar phenomena may also occur in the polar solution environment of the cell. When the polar solution is unbalanced in the cell of the seed, the cell may die, and the seed will lose its vigor [37].

Self-oxidation of seeds in storage environments with low water content exacerbates the process of lipid self-oxidation [38]. Lipid peroxidation can also lead to the accumulation of superoxide free radicals. As a reaction substrate, superoxide free radicals will be oxidized to $H_2O_2$. In fact, this increases the production of ROS. Several studies have shown that lipid autoxidation products—generally aldehydes and ketones—have been found in wheat and pea seeds [25,39]. In the aging research and analysis of corn seeds, it has been found that free fatty acids, such as linoleic acid and linolenic acid, will be oxidized and degraded into ketones, aldehydes, and alcohols in the process of lipid peroxidation. Several products, such as hexanal and nonanal, will have a negative impact on the vigor of seeds [40,41].

As a representative product of lipid peroxidation, MDA reduces the vigor of seeds by attacking membrane protein modification [42]. It was also found that the content of MDA increased with the decrease of seed vigor in other seeds [43]. Other studies on ROS-induced lipid peroxidation in plant seeds are described in the Table 1.

**Table 1.** Studies on lipid peroxidation, enzyme activity changes and gene damage during seed vigor decline.

| Physiological Alternation | Type of Seed | Method of Treating Seeds | Target | Method of Analysis | Performance | Ref. |
|---|---|---|---|---|---|---|
| Lipid peroxidation | Almond seeds of 4 varieties | NA | MDA | Spectrophotometry | MDA increase | [44] |
| | Bitter gourd | AA [a] | | | MDA increase | [45] |
| | Wheat | AA [a] | | | No significant change (0.38–0.45) | [25] |
| | Karanj | NA | | | Positive correlation with aging days (r = 0.96) | [46] |
| | Pea | Drying | | | MDA increase | [39] |
| | *Moringa oleifera* oilseed | AA [a] | | | MDA increase | [43] |
| | Spunta | Controlled environmental conditions | | | MDA increased | [47] |

<div align="center">

**Table 1.** *Cont.*

</div>

| Physiological Alternation | Type of Seed | Method of Treating Seeds | Target | Method of Analysis | Performance | Ref. |
|---|---|---|---|---|---|---|
| | *A. indicum* and *A. macrostachyum* | No special treatment | | | MDA increase | [48] |
| | *Brassica oleracea* L. and *Lactuca sativa* L. | Controlled deterioration | | | The change of cabbage seeds was not obvious, lettuce seeds increased significantly | [30] |
| | *Shorea robusta* | No special treatment | MDA; CD; HNE; FFA; LOOH; TL | Other: Spectrophotometry TL: Petroleum ether distillation | MDA: Positive correlation with aging days (r = 0.98) CD: Positive correlation with aging days (r = 0.92) HNE: Rose and then fell. FFA: Positive correlation with storage days (r = 0.96) LOOH: Positive correlation with storage days (r = 0.98) TL: Negatively correlated with storage days (r = −0.98) | [49] |
| | Pea | NA | POL | TCL | POL increased after seed death | [50] |
| | Rice | NA | Crotonaldehyde; H NE | HPLC-ESI-SIM | Increased | [51] |
| Enzyme activity anomalous | Rice | AA [a] | SOD APX CAT GR DHAR MDHAR | Spectrophotometry | APX, CAT, MDHAR: Decreased SOD, GR, DHAR: No significant change | [3] |
| | Beech | NA | CAT | | Decreased | [6] |
| | Oat | AA [a] | SOD APX MDHAR DHAR GR | | Decreased | [52] |
| | Karanj | NA | SOD CAT APX | | Decreased | [46] |
| | *Jatropha curcas* L. | AA [a] | SOD CAT POX | | Decreased after rise | [53] |
| | Tomato | AA [a] | CAT POD SOD APX GR DHAR | | Decreased | [54] |
| | Nonheading Chinese cabbage | AA [a] | PGI MDH | | Decreased after rise | [55] |
| | Bitter gourd | AA [a] | SOD CAT APX | | Decreased | [45] |
| | Wheat | AA [a] | SOD CAT GR | | Decreased | [25] |
| | Spunta | Controlled environmental conditions | SOD CAT GPX | | Relatively stable | [46] |
| | *Moringa oleifera* Lam. | NA | CAT SOD APX POX | | CAT, APX, SOD: Decreased POX: Increased | [21] |

<div align="center">**Table 1.** *Cont.*</div>

| Physiological Alternation | Type of Seed | Method of Treating Seeds | Target | Method of Analysis | Performance | Ref. |
|---|---|---|---|---|---|---|
| | *Brassica oleracea* L. and *Lactuca sativa* L. | Controlled deterioration treatment | CAT GR SOD α-amylase β−1,3-glucanase | α-amylase: dinitrosalicylic acid, β−1,3-glucanase: Spectrophotometry | Decreased | [30] |
| | Maize | NA and AA [b] | AOX1, ADH1, COXc, ATPase | Spectrophotometry | Upregulation of ADH1. Downregulation of COXc. ATPase and AOX1 is almost unchanged | [56] |
| | Oat | Controlled deterioration treatment | DEGs | | Downregulation of TCA and ETC related genes. Upregulation of antioxidant enzyme related DEG | [57] |
| | Pea | AA [b] | PsAPX, PsSOD, PsGRcyt, PsGRcm | | Decreased in the transcript abundance | [58] |
| | Rice | NA | OsAKR-1-OsAKR-3, OsALDH2-1-OsALDH2-5, OsALDH3-1-OsALDH3-5, OsALDH7 | qRT-PCR | Transcript decreased | [50] |
| Gene damage | | AA [a] | Cu/ZnSOD1-Cu/ZnSOD3, FeSOD, MnSOD, CAT1-CAT3, APX1-APX8GR1-GR3 DHAR1, MDHAR1, MDHAR2 | | Downregulation of Cu/ZnSOD family genes, CAT1, CAT2, APX1, APX4, APX5, APX6, GR2, DHAR1 and MDHAR1. Upregulation of APX3, APX7, APX8, GR3. | [3] |
| | Beech | NA | Genomic DNA | DNA extraction and electrophoresis | DNA laddering effect | [6] |
| | *Pisum sativum* | Drying | PsDHN2, PsDHN3, PsSBP65, PsHSP17.7, PsHSP18.1, PsHSP18.2, PsHSP22.7, Ps2-Cys prx | | Expression enhancement | [39] |
| | *Mung bean* | Controlled deterioration treatment | FSD1, MSD1, GST1, ZEP/ABA1, SDR, CPS1, SAM1, ACS | qRT-PCR | Downregulation of FSD1, SAM1. ACS. Upregulation of MSD1, ZEP/ABA1, SDR. | [59] |
| | *Brassica napus* L. | High temperature | DEGs | | 40 °C (83 upregulated and 37 downregulated). 60 °C (96 upregulated and 117 downregulated). 40/60 °C of heat stress exposure (88 upregulated and 158 downregulated) | [60] |
| | Rice | AA [a] | OLGLYI3 | | Decreased expression | [61] |

NA—Natural aging; AA [a]—Artificial aging; AA [b]—Accelerated aging.

### 2.3. Abnormal Protein Modification

According to Zavadskiy, et al. [62], protein hydroxylation refers to the irreversible structural damage of proteins caused by the oxidative modification of amino acids on proteins due to the high production of highly active oxidants during cell aging. During the aging process of plant seeds, ROS, MDA, and other products of lipid peroxidation, such as HNE, play a major role in protein carbonylation. It was found that the content of malondialdehyde in some seeds was positively correlated with the performance of protein carbonylation [34,63]. MDA will have Maillard-type reaction with protein and add hydroxyl into the amino acid side chain structure of protein [64]. In living cells, hydroxyl radicals generated by Fenton reaction (1) guided by metal catalytic oxidation (MCO) attack the main chain and amino acid side chain of protein to hydroxylate the protein [65]:

$$Fe^{2+}Cu^{2+} + H_2O_2 = Fe^{3+}Cu^{3+} + HO^- + HO^{\cdot} \tag{1}$$

Hydroxylated proteins will accumulate and aggregate, which will affect the cell vigor, such as the reduction of soluble protein content, the decrease of enzyme activity, the loosening of cell wall and so on. In the study on the aging of coix and beech seed, it was found that the content of soluble protein was negatively correlated with the degree of aging and positively correlated with the vigor of seeds [6,7]. According to another speculation of Agmon et al. [36], the loss of cell membrane characteristics may be related to the structural changes and functional impairment of proteins embedded in the cell membrane. The research results on mitochondrial membrane channel proteins of elm *(Ulmus pumila* L.) seeds also provide a basis for this hypothesis [34]. In short, seeds are rich in a variety of proteins. The irreversible oxidation of proteins by ROS and its indirect products will have an irreparable impact on the vigor of seeds.

### 2.4. Anomalous Enzyme Activity

One of the causes of seed aging is the decline or loss of enzyme activity in organisms, which is related to the attack of ROS and its indirect products. As the main role, enzymes play a positive role in seed growth. At present, the changes of enzyme activity during seed aging are mainly aimed at antioxidant enzymes and germination enzymes. Antioxidant enzymes are the main force to prevent ROS from attacking seed cells. Among them, SOD, which is commonly used to study and analyze the changes of enzyme activity during seed aging, will catalyze free radicals ($O^-$) to produce stable $O_2$ and $H_2O_2$, and then decompose into water and $O_2$ under the action of CAT or POD, so as to avoid the harm of free radicals to seed cells [66].

Due to improper storage methods (long storage time or serious dehydration in the storage environment), ROS will be produced and accumulated in excess. The dynamic balance between ROS and antioxidant enzymes is broken, the activity of antioxidant enzymes is reduced, oxidative stress occurs, and a large number of ROS outbreaks affect seed vigor [67]. The research on karanj *(Pongamia pinnata)* shows that the contents of $O^-$ and $H_2O_2$ are negatively correlated with seed water content and positively correlated with seed storage time. The activity of antioxidant enzymes is positively correlated with the water content of seeds [52]. Similar phenomena have also occurred in elm and beech seeds [6,34], but it is not particularly obvious in wheat seeds [25], indicating that the relationship between antioxidant enzyme activity and seed aging may also be related to species.

In the process of seed growth, the activity of germination enzyme also decreased in the process of seed aging. In the controlled deterioration treatment of cabbage and lettuce seeds, it was found that the content of α-amylase and β−1,3-glucanase decreased [30], and the loss of activity was also found in low-vigor rice seeds [68]. Other changes in enzyme activity are described in the Table 1.

### 2.5. Gene Damage

As a genetic structure, DNA will constantly change to meet the requirements of the environment. The outbreak of ROS will damage DNA and RNA. Hydroxyl radicals (· OH) react with base to produce free radical intermediates. At present, the most common form of damage is DNA damage caused by guanine (G) oxidation product 7,8-dihydro-8-oxo-guanine (8-oxo-G), which destroys the due principle of base complementarity [69]. This form of damage is expected to destroy 103 cells per day in normal tissues [70]. DNA methylation plays an important role in the process of seed development. Generally, methyl (CH3-) is added to the designated position of DNA, which usually occurs in the process of DNA replication and maintains the steady-state balance through DNA demethylation [71]. There is evidence that the outbreak of $H_2O_2$ in ROS will break this balance, which shows that 5-methylcytosine (5-mC) is oxidized to 5-hydroxymethylcytosine (5-hmC). This change tilts the above balance towards DNA methylation, resulting in DNA hypermethylation [8]. DNA hypermethylation can aggravate oxidative stress and accelerate apoptosis [72].

Evidence of DNA damage caused by ROS was found in the study of Brinjal (*Solanum melongena* L.) seeds. The results showed that when the content of ROS reached 11.84AU, the damage of the DNA tail was low (0.6%), and when the content of ROS reached 23.37AU, the damage of the DNA tail would reach 11.08%. In particular, some genes, such as OGG1 and PCNA, which repair DNA damage caused by 8-oxo-G or genes that regulate the expression of antioxidant enzymes such as CAT1 and ROS, show strong correlation or indirect strong correlation [73].

Recent studies have found that reactive carbonyl substances (RCS), an indirect product of ROS, such as crotonaldehyde, can damage RNA in rice seeds, which are mainly responsible for genes encoding carbonyl scavenging enzymes. The results showed that the expression ability of OsALDH2-2, OsALDH2-5, and OsAKR1 genes decreased by about 90%, 71%, and 43.5%, respectively, after aging treatment of rice seeds for about one year. The transcription levels of other ALDH family genes, such as OsALDH2-1 or OsALDH3-5, did not change significantly, and a negative correlation with seed vigor was even found in the analysis of OsALDH3-1. This may be due to the high resistance of these genes to ROS and its indirect products [50].

In addition, we found similar phenomena in studies on peas, Arabidopsis and beech [6,39,74]. In conclusion, the damage of ROS and its products to genes is fatal to seeds. During the storage of seeds, genes are the key to the seed's regulation and adaptation to the storage environment. Destructive gene damage will give incorrect instructions to the subsequent response of the seed to the environment and produce a series of continuous errors in the process of replication, transcription, and modification, which will eventually lead to the death of the seed. Several studies on gene expression during the decline of seed vigor are briefly introduced in Table 1.

### 2.6. Destruction of Organelle Structure

The explosion of ROS in stored seeds and the destruction of organelles are also one of the reasons for the loss of seed vigor. The main production site of ROS is mitochondria, which is responsible for cell respiration and energy supply for seeds. Therefore, it is also named "powerhouse". It is one of the organelles most sensitive to cell damage. When ROS accumulate in large quantities, mitochondria usually swell, aggregate, and suffer a collapse of the inner membrane or a blurred dissipation of mitochondrial outline; in severe cases, mitochondria may even be broken down [34,55]. This may be caused by ROS attacking mitochondrial membrane structure. Another interesting phenomenon was found in the study of pea seeds. There is a certain relationship between the integrity of mitochondrial inner membrane (IMM) and the vigor of seeds [75]. This conclusion resonates with the function of mitochondrial crest to a certain extent.

In addition to mitochondria, other organelles in seeds such as proteomes, adiposomes, and endoplasmic reticulum also changed due to the influence of ROS [55]. The nucleus is the largest and most important cell structure in eukaryotic cells. It is the main place for the

storage, replication, and transcription of genetic information in cells. According to recent studies, ROS can cause disorder or even destruction of the morphology of the nucleus and nuclear membrane, obvious matrix cavity, and chromosome breakage [76–78]. This may be caused by the damage and degradation of cell membrane and cell wall and the massive outbreak of ROS attacking the nucleolus.

Various organelles in seeds are tools to maintain the vigor of seeds during storage. The burst of ROS will damage various organelles, affect the working efficiency of organelles, and finally bring irreparable decline of vigor to seeds. However, at present, the interaction between various organelles affected by ROS is not clear, and the details of the competitive relationship between self-healing and passive destruction of various organelles after being attacked by ROS are unknown. We also hope that more scholars can discuss these in this regard in the future.

In conclusion, the decline of seed vigor is closely related to the changes of chemical composition and microstructure in seeds, such as lipid decomposition, protein carbonylation and mitochondrial structure changes. These changes provide a theoretical basis for the existing nondestructive testing of seed vigor. At present, these microchanges of seeds during storage can be detected by nondestructive methods, which can be used for qualitative and quantitative analysis of the vigor of a large-scale seed. On the other hand, understanding the seed vigor under various physiological conditions and chemical composition levels will also help us to further clarify the principle of the biological changes of seed-vigor decline. We will discuss these in the following sections.

## 3. The Relationship between Spectrum and Chemical

Molecular spectrum test technology is a method that reflects the structural information inside the molecule through the spectrum generated by the transition between molecular vibration energy levels or rotational energy levels, determines the molecular moment of inertia, molecular bond length, bond strength, and dissociation energy, and finally realizes the qualitative and quantitative test of chemical composition information in the target. Common spectral analysis technology, including near-infrared spectroscopy (NIR), visible near-infrared spectroscopy (VIS-NIR), short wave infrared spectroscopy (SWIR), and Raman spectroscopy, has been widely used in the research for analyzing agricultural products and testing seed vigor because it has the advantages of causing no damage to the tested object, requires only a fast test, carries low risk, and is environmentally friendly.

In the spectrum, different chemical composition in the sample will appear in different bands. In the common studies of analyzing seed vigor by spectrum, chemicals including proteins, carbohydrates, and lipids, which ensure the normal operation of seed life activities, are mainly analyzed. The characteristic groups of these chemical composition ($-CH_3$, -COOH, -OH, amides, and carbonyl), will leave marks in the spectrum. For example, amides will show characteristics in the NIR spectrum of 7400–6540 $cm^{-1}$ (molar absorptivity 0.7), 5160–5060 $cm^{-1}$ (molar absorptivity 3.0), 5040–4990 $cm^{-1}$ (molar absorptivity 0.5), which is related to the overtone of N-H stretch, second overtone of C = O stretch, second overtone of N-H deformation, and combination of C = O and N-H. Refer to section III of *Lange's Handbook of Chemistry (16th Edition)* [79] for more details.

As for the close relationship between seed vigor and spectrum, several research papers have reported the quantitative detection of seed vigor by using spectroscopy technology. Changes caused by seed aging (ROS accumulation, etc.) will affect the optical properties of seeds [80]. Fan et al. [9] used SPA to select nine optimal wavelengths between 1265 and 2168 nm to obtain a vigor classification model. These spectral regions are respectively related to the overtones generated by stretching vibration at C-H, O-H, N-H, and C = O, and represent the changes of protein, starch, and water. In the study of heat-damaged and artificially treated seeds, Wang et al. [81] showed absorption peaks at 1000, 1200, 1400, and 1700 nm on the original spectra, which respectively represent the content of carbohydrate, lipid and protein in corn seeds. In addition, the reflectance of the treated seeds in the spectrum is also significantly different from that of the original seeds, which indicates the

change of chemical components in the seeds during treatment. Low temperature will cause forest damage to seeds, which will lead to cytoplasmic dehydration and protein damage. In the study of detecting frozen seeds, Jia et al. [82] found that the two spectral regions of 1500 to 1700 nm and 1800 to 2100 nm in the spectral have major contributions to the vigor test model. These regions are related to the overtones produced by C-H, O-H, and N-H, and indicate the changes of chemical substances, such as proteins. Satisfying results were also obtained in the experiment of identifying normal seeds and freeze-damaged seeds by the above model. Other studies on spectral characteristics and corresponding compounds in seed vigor analysis are shown in Table 2.

**Table 2.** Studies on spectral characteristics and corresponding compounds in seed vigor analysis.

| Type of Spectral | Material | Object | Band | Remarks | Ref. |
|---|---|---|---|---|---|
| NIR | Wheat | Protein, water, and starch | 1200–2400 nm | Peak 1 (1265 nm, 1272 nm and 1278 nm,): Related to the starch and protein. Peak 2 (1912 nm and 2168 nm): Related to the protein. Peak 3 (2027 nm): Related to the starch. | [9] |
| NIR | Maize | Protein, carbohydrate, and water | 1000–2400 nm | Peak 1 (1180 nm and 1420 nm): Second overtone of C-H stretching. Peak 2 (1700 nm and 1748 nm): Overtone of C-H stretching. Peak 3 (1918 nm): Related to the carbohydrate. Peak 4 (2035 nm): Related to the protein, water and starch. Peak 5 (2058 nm): Overtone of N-H stretching. Peak 6 (2275 nm): Overtone of O-H stretching. | [83] |
| SWIR | Maize | Starch, lipid, cellulose, protein | 1100–1600 nm | Peak 1 (1220–1230 nm): Second overtone of C-H stretching (starch, lipid, cellulose). Peak 2 (1560–1570 nm): First overtone of N-H stretching (protein). | [84] |
| NIR-HSI | Rice | Protein | 900–1700 nm | Peak 1 (1000 nm): Second overtone of N-H stretching (protein). Peak 2 (1200 nm): Second overtone of C-H stretching. Peak 3 (1400 nm): First overtone of O-H stretching. | [85] |
| HSI | Maize | Starch. Lipid, protein, cellulose, sugar | 1000–2200 nm | Peak 1 (1100–1300 nm): Second overtone of C-H stretching (starch). Peak 2 (1450 nm): First overtone of O-H stretching (starch and lipid). Peak 3 (1700–1750 nm): Second overtone of C-H stretching (protein, starch, and lipid). Peak 4 (1800–2250 nm): Related to the protein, cellulose, and sugar. | [86] |
| HSI | Wheat | Protein, starch, lipid | 949–1638 nm | Peak 1 (1030.9 nm and 1047.9 nm): Third overtone of N-H stretching (protein). Peak 2 (1152.4 nm and 1334.9 nm): Second overtone of C-H stretching (starch). Peak 3 (1413.3 nm and 1529.6 nm): First overtone of O-H stretching (starch and ipid). | [87] |

**Table 2.** *Cont.*

| Type of Spectral | Material | Object | Band | Remarks | Ref. |
|---|---|---|---|---|---|
| Raman | Maize | Protein, starch, lipid, cellulose | 170–3200 cm$^{-1}$ | Peak 1 (964 cm$^{-1}$): Related to the cellulose. Peak 2 (1660 cm$^{-1}$): Related to the protein. Peak 3 (2800–3000 cm$^{-1}$): Related to the lipid. | [83] |
| Raman | Pepper | Protein, carbohydrate | 150–1800 cm$^{-1}$ | Peak 1 (1090 cm$^{-1}$): Overtone of -CO stretching. Peak 2 (1154 cm$^{-1}$): Overtone of C-C stretching. Peak 3 (1263 cm$^{-1}$): Overtone of =CH stretching. Peak 4 (1440 cm$^{-1}$): Overtone of =CH$_2$ stretching. Peak 5 (1520 cm$^{-1}$): Overtone of C=C stretching. | [88] |
| FI-NIR | Maize | Protein, starch, and carbohydrate | 1110–2500 nm | Peak 1 (1500–1700 nm) and Peak 2 (1800–2100 nm): Overtone of C-H, O-H and N-H stretching. | [82] |
| Vis-NIR | Maize | Protein, starch, lipid | 500–1100 nm. 1000–1850 nm | Peak 1 (750 nm): Third overtone of O-H stretching. Peak 2 (800 nm): Third overtone of N-H stretching. Peak 3 (900 nm): Third overtone of C-H stretching. Peak 4 (1100 nm): Second overtone of C-H stretching. | [81] |

## 4. Current Testing Methods for Seed Vigor

Testing seed vigor is the key to ensure that high-quality seeds flow into agricultural production and plays a vital role in the complete chain of modern agricultural production. Early seed-vigor test methods include the standard germination test, conductivity test, and TTC staining. The standard germination test is a method by which to directly simulate the germination behavior of seeds to test the vigor. During seed germination, the germination performance of seeds (such as length, abnormal rate and germination rate) is tested to analyze the vigor of seeds. This usually takes at least a week. Germination rate is a phenotypic trait that describes seed viability. It is the result of comprehensive expression of many genes and a series of environmental effects. However, at the biochemical and molecular levels, it is difficult to find a biomarker that has a correlation with or can respond to the germination rate. The conductivity test is a method by which to test seed vigor by testing the electrolyte of seed leaching solution. It was first proposed in 1925 and developed to mature use in 1976 [89]. The principle of the conductivity test is related to the damage of the cell membrane, because the cell membrane of low-vigor seed is damaged to a large extent and difficult to repair itself. High-vigor seeds are the opposite. Indigo carmine is used early to detect seed vigor; live seeds are usually not stained, whereas dead seeds produce a very strong spot [90]. Similar to indigo carmine, TTC staining is also a test method for seed vigor. It was listed in the *International Rules for Seed Testing* by the International Seed Testing Association (ISTA) as early as 1953. It is a recognized simple and effective method for vigor testing. Generally, it takes tens of minutes or one day (depending on the seed type and the environmental conditions of the experiment) [34,91,92]. The principle is that the seed dehydrogenase is positively correlated with the seed vigor. This enzyme can reduce TTC to red 1,3,5-triphenylformazan (TPF), so as to intuitively judge whether the seeds are alive. It is reported that malate dehydrogenase (MDH), cytochrome c reductase, succinate dehydrogenase (SDH), and some enzymes involved in cellular information transmission system may be members of TTC reductase family [93]. The specific names of these enzymes



have not been completely checked in the current research. In a word, as an internationally recognized method, TTC staining has the advantages of simplicity and rapidity (compared with standard germination experiments), but it is not a method that meets the current requirements of large-scale seed-vigor tests because of its destructive effect to seeds. In the end, it is necessary to find a nondestructive method by which to judge whether a single seed is alive or dead. With the progress of current science, some optical methods that are faster, more convenient, and nondestructive have emerged rapidly in place of conventional methods.

### 4.1. Vibrational Spectroscopic Techniques

As early as 1978, American scholars analyzed the water content of single corn seeds by the NIRS method [94]. Subsequently, studies on the use of NIR to analyze the information of useful components, such as oil content, starch content, and the protein quality of corn seeds also appeared one after another [95]. This laid a solid theoretical foundation for the later study of seed vigor.

In recent years, more studies emerged on the establishment of the model of seed-vigor identification through vibration spectrum technology combined with statistical analysis theory, and the model accuracy of the seed-vigor test also achieved gratifying results, especially in screening the seeds that have been damaged by heat or have lost their vigor after artificial aging treatment. In 2012, Esteve Agelet et al. [96] used NIRS to test the vigor of heat-damaged seeds by combining four statistical models: partial least squares analysis (PLS-DA), similarity classification (SIMCA), minimum support vector machine (LS-SVM), and K-NN, with the highest accuracy being 99%. Subsequently, Ambrose et al. [83] used FI-NIR, and Raman spectroscopy combined with the PLS-DA model. In the test of white corn, yellow corn, and purple corn, the accuracy of the training set and verification set was more than 95% (Figure 2B). With the maturity of spectral analysis technology and the gradual improvement of statistical models, recent similar studies have also produced results with similar conclusions or higher accuracy [9,81,97,98].

Furthermore, the research on the analysis of seed vigor by vibration spectrum analysis technology combined with visual technology has also emerged. Among them, the commonly used visual technology is hyperspectral imaging technology (HSI). Generally, the image with spectral resolution of $10^{-2}\lambda$ is called hyperspectral imaging technology, which is a new way to perfectly combine the traditional two-dimensional technology and spectral analysis technology. It not only has the characteristics of fast, efficient, and real-time analysis of traditional spectral technology, but also has the advantages of intuitive image, good reproducibility, and high flexibility. It realizes the comprehensive analysis of samples from multiple angles. HSI has appeared in the test of seed vigor for the first time since 2016. Ambrose et al. [17] obtained more than a 95.6% accuracy in the training set and verification set of white corn with vis NIR-HSI in the wavelength range of 400–1000 nm and SWIR-HSI in the wavelength range of 1000–2500 nm combined with PLS-DA model, but the results obtained in the test of yellow corn and purple corn are not satisfactory. With the continuous development of relevant technologies, there has been some research on the analysis of seeds combined with deep learning technology (DL), including work on convolutional neural networks and the continuous projection algorithm [18,99,100], which has achieved some exciting results. The classification accuracy has reached 83.3% [83], 100% [84] (Figure 2C), and 99.35% [86] (Figure 2A) in corn seeds, and 95% [1], 99.2% [98], and 99.93% [87] in other seeds. Table 3 shows some studies using spectroscopy or spectral imaging technology in the seed-vigor test.

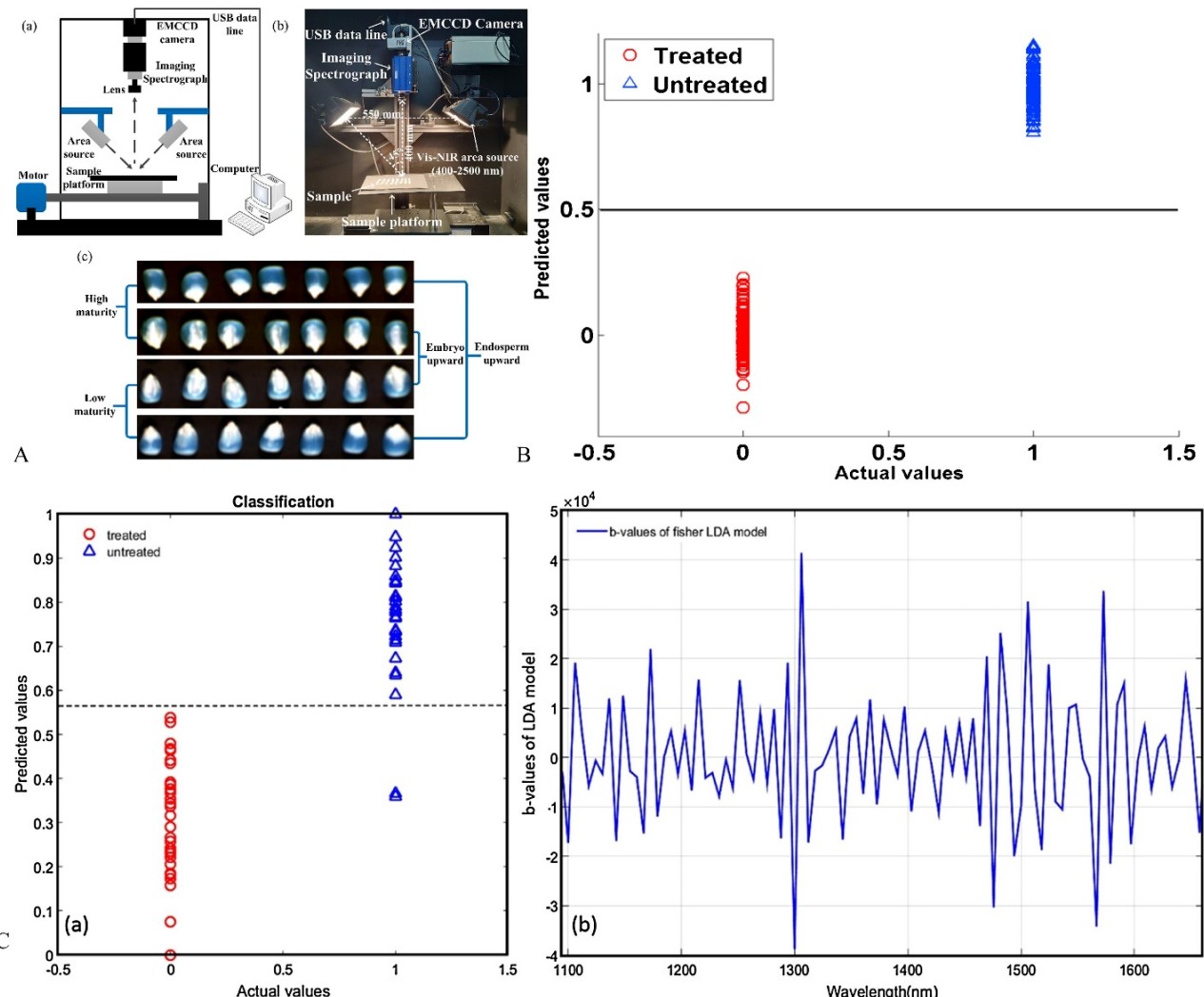

**Figure 2.** (**A**). NIR-HSI acquisition equipment of Wang, et al. (**B**). Ambrose, et al. used FT-NIR spectra of PLS-DA model to classify maize seeds. (**C**). Classification (**a**) and β-values (**b**) of LDA model by Wakholi, et al.

**Table 3.** Application and results of spectral analysis and related technology in seed-vigor research.

| Type of Seed | Preparation | Spectral Analysis Technology | Analysis Model | Classification Accuracy | Ref. |
|---|---|---|---|---|---|
| Maize | Microwave treatment | NIR | PLS-DA | 100% | [97] |
| | Artificial aging | NIR. Vis-NIR | PLS-DA; (500–1100 nm and 1000–1800 nm) CARS-PLS-DA | 98.7% | [81] |
| Maize (white, yellow, and purple corn) | Microwave treatment | FI-NIR. Raman | PLS-DA | 100% | [83] |
| | | Vis-NIR HSI. SWIR HSI | PLS-DA | 97.6% | [17] |

**Table 3.** *Cont.*

| Type of Seed | Preparation | Spectral Analysis Technology | Analysis Model | Classification Accuracy | Ref. |
|---|---|---|---|---|---|
| Corn | Microwave treatment and frost treatment | NIR | PLS-DA. SIMCA. LS-SVM. K-NN | 99.0% | [96] |
| Maize | Frost treatment | NIR | PCA-OLDA or PLS-OLDA to extract spectral feature SVM, BPR, MD | 99% | [82] |
| Corn (yellow dent corn, white dent corn, and purple flint corn) | Microwave treatment | SWIR-HSI | LDA, PLS-DA, SVM | 100% | [84] |
| Maize | Artificial aging | ATR-FITR | PLS-DA | 85% | [41] |
| Maize (Huanong 101) | Artificial aging | HSI | SVM. ELM. DCNN | 100% | [101] |
| Maize (Deyu 977) | Frost damage | Vis-NIR-HSI | KNN. SVM. ELM. DCNN | 100% | [102] |
| Maize1 (101101) and Mazie2 (7879) | Artificial aging | HSI | SVM | 100% | [103] |
| Maize (Zhengdan 958) | Maturity division | NIR-HSI | PLS-DA. DT. AddBoost | 99.35% | [86] |
| Retinispora (Hinoki cypress) | Not mentioned (may be natural mold infection) | FI-NIR | PLS-DA. VIP-PLS-DA | 99.2% | [98] |
| Japanese mustard spinach (Brassica rapa var. perviridis) | Storage for one year | NIR-HSI | CNN | - | [99] |
| Wheat (Luyuan 502) | Artificial aging | Vis-NIR-HSI. SWIR-HSI | PLS-R. SPA-PLS-R. PLS-R-RC | 94.8% | [1] |
| Rice (Zhongzao39, Zhongzao 8, Y liangyou2) | Storage for 5–7 years | NIR-HSI | LR; SVM; CNN; PCA-LR; PCA-SVM; PCA-CNN | 86.67% | [85] |
| Rice (Yanfeng) | Frost damage | NIR-HSI | DT; KNN; SVM; DF | 99.93% | [87] |
| Wheat (Xinong 979, Xinong 20, Zhengmai 366, Xinmai 26, Zhoumai 26 and Chinese Spring) | Artificial aging | NIR | PCA/SPA + SVM/ELM/RF/AdaBoost | 92.2% | [9] |

The molecular vibration spectroscopy technology has made rapid development in agricultural research, especially in monitoring seed vigor, and achieved remarkable results. However, molecular spectroscopy technology still has some limitations. For example, it is difficult to achieve high-precision analysis in large-scale tests of NIR-HSI [99]. The obtained model is generally only applicable to the studied species, and more research is needed for other members of the same family [98]. The FI-NIR spectrum needs a long detection

time and a large number of manual operations for single-seed analysis, which limits its prospects for use on an industrial scale [83]. In addition, the detection temperature, sample properties, system vibration, and imperfect lighting may produce noise in the process of data collection, affecting the accuracy of the model [84]. With the development of optics, computer science, data processing technology, information transmission technology, and other related technologies, we believe that molecular spectral vibration technology will bring us more accurate and efficient performance in the future.

*4.2. Biological Speckle Laser*

In 1975, when the British scholar Dainty [104] used lasers to irradiate objects, he found a surprising phenomenon: when one irradiates objects with highly coherent lasers, one will get an image composed of special particles with chaotic arrangement and irregular combination. The phenomenon formed in this way is called "laser speckle". In essence, the laser speckle belongs to an optical imaging technology. Therefore, it also has the advantages of noncontact, no damage and fast image generation. However, due to its chaotic imaging, the results can only be described by statistical theory. Since its discovery, laser speckle, especially the biospeckle laser, has been effectively used in the test of fungal infection during seed storage [105], damage during seed germination [106], identification of fruit or vegetable frostbite [107], analysis, and understanding of human erythrocyte dynamics [108].

In the test of seed vigor, researchers have observed the biospeckle phenomenon caused by various physiological processes in seeds [109]. In 2003, Braga, et al. [110] prepared viable and nonviable bean seeds in advance, and successfully found their difference by using the biospeckle laser. In 2005, some scholars bisected corn seeds along the embryo axis and analyzed the seed vigor through biospeckle laser phenomenon [111]. In order to make the obtained image more direct, the concept of wavelet entropy (WE) was introduced to grade the gray area of the image according to the weight to obtain the high-activity area and low activity area of the seed. Thus, the image is reasonably explained and described. One explanation for this approach is that the presence of water in nonviable tissue. Finally, the TTC staining method commonly used in agricultural engineering is used to verify the rationality of WE-BSL method. The advantage of biospeckle laser is that the results of TTC staining for 24–48 h can be obtained in approximately one hour, which greatly improves the efficiency. With the updating of related technologies, researchers have found a faster method. Using fuzzy granularity to analyze the activity distribution area of seeds can be realized in only tens of milliseconds [112]. Then, Braga et al. [113] analyzed the homogeneous region of maize seed bisection through the method proposed by Fuji. They calculated the normalized symbiosis matrix constructed by time history speckle pattern (THSP) by using Formula (2), and finally numerically described the image by using the moment of inertia calculated by Formula (3), obtaining the same conclusion as previous scholars, and confirming the theory of biospeckle laser test of seed vigor. We have

$$COM_{ij} = \frac{COM_{ij}}{\sum_i N_{ij}} \tag{2}$$

$$IM = \sum_{ij} COM_{ij}(i-j)^2 \tag{3}$$

In terms of screening seed vigor, biospeckle laser seems to still be in its infancy, but encouraging results have been obtained [109,114,115], especially the impact of seed-tissue moisture on the measurement results; that is, whether the initial water content of seeds will affect seed germination [116]. This kind of research undoubtedly lays a foundation for the test of seed vigor by biospeckle laser, because in the process of seedling raising, the content of initial water will participate in part of the physiological process of seed germination. In recent years, several consecutive studies have shown good performance, which shows that the biospeckle laser has great applicability in testing seed vigor. Table 4 shows the

details and conclusions of several studies on the application of biospeckle laser in testing seed vigor.

**Table 4.** Performance of biospeckle laser and other method in seed-vigor research.

| Method | Type of Seed | Preparation | Statistical Theory | Performance | Ref. |
|---|---|---|---|---|---|
| Biospeckle laser | Bean | Control moisture content. Prepare viable and nonviable seeds | COM-IM. GD; | Speckle can be used to distinguish the vigor of seeds | [110] |
| | Maize | Imbibition and bisection | WE | The distribution area of maize seed vigor was found | [111] |
| | Maize | | Gr | | [112] |
| | Maize | Bisection | IM | | [113] |
| | Maize and bean | Maize: Not mentioned Bean: dead and live seeds with different water content | Fuji; Entropy; IM | The distribution area of maize seed vigor was found and different manifestations of obtaining dead seeds and live seeds | [117] |
| | Coffee bean (*Coffea arabica* L.) | Not mentioned | STD biospeckle index | Accuracy: 87.5% | [115] |
| | Chickpea (*Cicer arietinum* L.) | Water treatment and chemical treatment | MSF-BA | BA and GR were positively correlated (R = 0.88, $p < 0.01$) and negatively correlated with MGT (r = −0.92, $p < 0.01$) | [109] |
| | Bean | Disinfect and dry | FTHSP + COM + AVD | BA was positively correlated with the initial water content of seeds (0.97–0.99) and related to seed germination. | [116] |
| | Soybean [*Glycine max* (L.) Merrill] | Imbibition | CNN; TL (VGG-16, VGG-19, InceptionV3, ResNet50) | Accuracy: 98.31% | [114] |
| X-ray CT imaging | *Moringa peregrina, Abruspreca torius, Acacia tortilis, Acacia ehrenbergiana* and *Arthrocnemum macrostachyum* | No special treatment | - | High vigor seeds have clear morphology Low-vigor seeds have morphology damage | [118] |
| X-ray | Muskmelon seed | | LDA | Accuracy: 98.9% | [119] |
| X-ray CT imaging | *Crambe abyssinica* | | CNN | Accuracy: 95% | [19] |
| Biosensor (MCLA) | Pepper | Natural aging | PLS-DA | Accuracy: 88.7% | [120] |
| EIS | Rice (Jing Dao No. 21), Maize (Tai Gu No. 1/2), Wheat (Ze Yu No. 2) | Artificial aging | Chemiluminescence induced by MCLA | Positively correlated with TTC staining (r = 0.99) | [121] |
| Volatile metabolites (ethanol) | Rice | Natural aging | FLD | Accuracy: 90% | [20] |
| GC-IMS | Melon seed | - | | Significant correlation with other germination indexes | [122] |

**Table 4.** *Cont.*

| Method | Type of Seed | Preparation | Statistical Theory | Performance | Ref. |
|---|---|---|---|---|---|
| X-ray CT imaging | Sweet corn | Natural aging and Artificial aging | GC-IMS-AS-PCC-VIP-PLS-R | Accuracy: 94.7% | [40] |
| IDS | *Juniperus polycarpos* seed | No special treatment | - | Germination rate has improved | [123] |
| IDS | *Pinus patula* seeds | | | | [124] |
| IDS | *Schinus molle* L. seeds | | | | [125] |
| Atmospheric Pressure Plasma Treatment | *Capsicum annuum* L. and *Trichosanthes cucumerina* seeds. | Stored at 25 °C for one year | - | Germination rate has improved | [126] |

Biospeckle laser has become a popular diagnostic tool in the 21st century because of its simple experiment, fast acquisition, relatively stable analysis process, relatively simple algorithm. However, there is still little research on seed vigor, and the research on seed types is not extensive enough. More types of seed need to be verified by experiments. At present, scholars have made efforts to explore commercialization in this area. We also believe that this emerging method for testing seed vigor may become a possibility to test the vigor of large quantities and all types of seeds in the near future.

*4.3. Other Novel Means*

In addition to the above common methods for evaluating seed quality, in recent years, seed experts have also demonstrated several other methods by which to test seed vigor. These methods also have the characteristics of being nondestructive, requiring a fast test, having strong objectivity, reducing human subjective influence, and being highly accurate and stable.

X-ray is a technology used to evaluate internal physical information, which has been widely used in the evaluation of seed vigor in previous years. Viable seeds have complete internal structure, a smooth seed coat, and regular endosperm shape, whereas dead seeds have an internal cavity, an endosperm deformity, seed coat damage, and so on [127]. According to these standards, an X-ray can judge whether the seeds still have the significance of putting into agriculture through the collected images [118]. Recently, the evaluation of seeds by X-ray combined with machine learning, deep learning, and a statistical model has achieved good accuracy, which proves that this method is a very promising technology in seed research [19,120].

Satisfactory results have been obtained in the test of mechanical damage, or terminal physiological death of seed. However, the disadvantage of the X-ray is also obvious. This method seems to only get the judgment of whether a seed is "alive" or "dead" because it is limited to expressing the current physical state of seeds, not microscopic chemical damage or physiological changes [119]. It is necessary to further improve the ability of X-rays to distinguish the microstructure and study the correlation between the change of microstructure and seed vigor.

Recently, analyzing the vigor of seeds according to the gas produced by seeds has become a method of great concern. This method has been used by the people for a long time. For example, when buying rice or beans, some consumers will pick up a handful and smell them to decide whether to buy or not. The carbohydrates and lipids of the seeds affected by aging will be oxidized and decomposed into odorous aldehydes and ketones, which usually have an unpleasant sour smell [128]. Compared with ordinary consumers, what seed scientists need to do is to analyze seeds that cannot be used for agricultural production in the early stage of seed vigor decline. At present, this method has been effective in the analysis of rice, corn, and melon seeds, and is expected to become a standardized measurement tool for testing seed vigor in the future [40,122,128].

Incubation, drying, and separation (IDS) is also recognized as an effective sorting technique for extracting live seeds. The principle of this technique is that seeds that have lost vigor lose water more quickly during incubation and are therefore separated. It has been reported with good results in the seed sorting of the *Juniperus polycarpos* seed (K. Koch) [123], *Pinus patula* seeds [124], and *Schinus molle* L. seeds [125]. However, the efficiency of this method seems to be dependent on the empirical nature of the researcher and the species of the seeds. Sufficient experience of the researcher is needed to set the optimal conditions for IDS. There are other methods, such as biosensor and electrical impedance spectroscopy [20,121]. However, these are not used frequently, and these types of seeds are not used widely enough. Table 4 introduces all other emerging seed test methods.

## 5. Discussion and Conclusions

Seed vigor is not only an important index by which to evaluate seed quality and production value, but also the main basis for seed classification. During the process of seed harvesting, storage, and transportation, the seed vigor will decline slowly. Developing a single seed-vigor monitoring and early warning method is a worldwide problem because the decline of seed vigor is a slow and difficult physiological activity to be observed in real time. Many studies have shown that ROS plays a vital role in the change of seed vigor. A series of physiological activities that reduce seed vigor are related to the outbreak of ROS. In fact, the seed can resist ROS attack, but it can only deal with the normal content of ROS. The excessive accumulation of ROS leads to oxidative stress, and the balance between the antioxidant system and oxidant is broken, resulting in damage to the membrane structure, nutrients, and genes of seeds. As we all know, genes are the premise for seeds to maintain normal physiological activities and the brain that regulates seeds to adapt to environmental changes. The impact of ROS on genes will make seeds misjudge the current physiological state and directly cause the death of seeds. Mitochondria are the breathing places of cells, and endoplasmic reticulum is an important synthesis base of proteins, lipids, and carbohydrate in cells. Under the influence of ROS, these organelles are also damaged. Other studies have also found that the antioxidant system will gradually collapse at this time, which makes the balance between the antioxidant system and oxidants more inclined to oxidants, and the damage to seeds is undoubtedly fatal. This series of changes are all closely related to ROS. The current researchers have proven the influence status of ROS in seed aging, but the sequence of these changes seems to be unclear. Whether the organelles in the cell have the behavior of self-defense is not clear nor is the answer to the question of how we can compensate the seeds according to the physiological changes during the decline of seed vigor, so as to prolong the seed vigor as much as possible. These issues may need require more in-depth research from scholars.

At present, in terms of seed-vigor testing, the traditional methods have the advantages of stability and simple operation. However, the result is a conclusion based on a large amount of data. The traditional methods are only for applicable random inspection other than large-scale inspection. NIR, HSI, biological speckle, X-ray CT images, and other methods have the characteristics of causing no damage, not being time-consuming, and possessing real-time analysis traits that traditional methods do not have; furthermore, all of them have achieved satisfactory results. Among these methods, vibrational spectroscopy combined with visual techniques to detect seed vigor seems to be the most effective, because it is not only fast, efficient, and provides real-time analysis, but it also allows one to analyze the changes of chemical substances inside the seeds compared to other methods, thus accurately detecting seed vigor. However, they still have limitations. NIR and HSI are difficult to establish and maintain the model, which will be unstable when mutually using different batches and types of seeds. Biospeckle has not been reported to be used among a large number of seeds, and X-ray CT images are only suitable for physical damage or good results at the end of seed-vigor decline. If a connection between the nondestructive detection and the ROS content in seeds during storage can be established, it

will more effectively help us understand the changes of seed vigor. Thus, combined with the physiological changes of seed vigor decline, it is of great significance to explore a rapid, nondestructive, and stable single seed-vigor test technology and apply it to agricultural production. The commercial use of this technology is also the duties of the development of seed science in the future.

The sufficient supply of crops is very important under the background of epidemic infestation and population growth. Seeds are at the core of agricultural production and the carriers of inherited crop resources. Seed vigor is the standard by which to evaluate seed quality. We hope that by summarizing the physiological changes in the process of seed-vigor decline and the test methods of seed vigor, we can provide seed scientists with a nondestructive, reliable, and stable seed-vigor test mechanism and finally develop a reasonable and effective early warning mechanism of seed vigor decline.

**Author Contributions:** Conceptualization, M.X. and Y.L.; methodology, W.H.; software, Q.W.; formal analysis, X.T.; investigation, S.F.; resources, C.Z.; writing—original draft preparation, M.X.; writing—review and editing, Y.L. and W.H.; supervision, W.H.; project administration, Y.L.; funding acquisition, W.H. All authors have read and agreed to the published version of the manuscript.

**Funding:** This research was funded by the financial support of China Agriculture Research System (Project No. CARS-25-07) and The APC was funded by Project No. CARS-25-07.

**Institutional Review Board Statement:** Not applicable.

**Informed Consent Statement:** Not applicable.

**Data Availability Statement:** Not applicable.

**Acknowledgments:** The authors would like to thank the National Natural Science Foundation of China (31871523), the financial support of China Agriculture Research System (Project No. CARS-25-07) and Young Elite Scientists Sponsorship Program by CAST (2019QNRC001).

**Conflicts of Interest:** The authors declare no conflict of interest.

## Nomenclature

| | |
|---|---|
| ROS | Reactive Oxygen Species |
| HSI | Hyperspectral Imaging Technology |
| TTC | 2,3,5-Triphenyltetrazolium Chloride |
| MDA | Malondialdehyde |
| HNE | 4-Hydroxy-Non-2-Enal |
| NA | Natural Aging |
| AA[a] | Artificial Aging |
| AA[b] | Accelerated Aging |
| CD | Conjugated Diene |
| FFA | Free Fatty Acid |
| LOOH | Lipid Hydroperoxide |
| TL | Total Lipid |
| POL | Product Of Lipid |
| CAT | Catalase |
| SOD | Superoxide Dismutase |
| APX | Ascorbic Acid Peroxidase |
| GR | Glutathione Reductase |
| DHAR | Dehydroascorbic Acid Reductase |
| MDHAR | Monodehydroascorbate Reductase |
| GPX | Glutathione Peroxidase |
| POX | Peroxidase |
| PGI | Phosphohexose Isomerase |
| MDH | Malate Dehydrogenase |
| qRT-PCR | Quantitative Real-Time Polymerase Chain Reaction |
| TCL | Thermochemilumenescence |

| | |
|---|---|
| HPLC-ESI-SIM | High Performance Liquid Chromatography-Electrospray Ionization-Selective Ion Monitoring. |
| DEG | Differentially Expressed Genes |
| NIR | Near Infrared Spectroscopy |
| VIS-NIR | Visible Near-Infrared Spectroscopy |
| SWIR | Short Wave Infrared Spectroscopy |
| ISTA | International Rules for Seed Testing by the International Seed Testing Association |
| TPF | 1,3,5-triphenylformazan |
| PLS-DA | Partial Least Squares Analysis |
| SIMCA | Similarity Classification |
| LS-SVM | Minimum Support Vector Machine |
| DL | Deep Learning Technology |
| K-NN | K-Nearest Neighbors |
| FI-NIR | Fourier Transform Infrared |
| CARS | Competitive Adaptive Reweighted Sampling |
| PLS-OLDA | Partial Least Squares-Orthogonal Linear Discriminant Analysis |
| SVM | Support Vector Machine |
| BPR | Biomimetic Pattern Recognition |
| MD | Mahalanobis Distance |
| LDA | Linear Discriminant Analysis |
| ATR-FITR | Attenuated Total Reflectance-Fourier Transform Infrared Spectroscopy |
| ELM | Extreme Learning Machine |
| DCNN | Deep Convolutional Neural Networks |
| DT | Decision Tree |
| VIP | Variable Importance of Projection |
| CNN | Convolutional Neural Networks |
| PLS-R | Partial Least Squares Regression |
| SPA | Successive Projection Algorithm |
| RC | Regression Coefficients |
| LR | Logistic Regression |
| DF | Deep Forests |
| RF | Random Forest |
| COM-IM | Moment of Inertia of Co-occurrence Matrix |
| GD | Generalized Differences |
| WE | Wavelet Entropy |
| Gr | Granularity |
| STD | Standard Deviation |
| MSF-BA | Modified structural Function-Biospeckle Activity |
| FTHSP | Full-Field Time History of Speckle Pattern |
| AVD | Absolute Value of Difference |
| TL | Transfer Learning |
| MCLA | 2-methyl-6-(p-methoxyphenyl)-3,7-dihydroimidazo [1,2$\alpha$] pyrazin-3-one |
| ELS | Electrical Impedance Spectroscopy |
| GC-IMS | Gas Chromatography-Ion Mobility Spectrometry. |
| IDS | Incubation, Drying and Separation |

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
