# Peer review of "Physiological Alterations and Nondestructive Test Methods of Crop Seed Vigor: A Comprehensive Review"

_agriculture, doi:10.3390/agriculture13030527_

Round 1

Reviewer 1 Report

13.02.2023

Manuscript ID: agriculture- -2225672
Title: Physiological alterations and nondestructive test methods of crops seed vigor : A comprehensive review

Comments

The manuscript sent for evaluation contains 18 pages of text with tables, figures and 6 pages of reference list (124 references). From the last two years 2021 and 2022 it is over 27%.

Abstract: Complete, In the abstract, authors should list the most commonly used methods for assessing seed vigor and give an example of new ones.

Keyword: The keywords should not contain repeated words from the title of the article: remove – “seed” and “vigor”

Many abbreviations are used in the text of the work, so I suggest introducing a list of abbreviations at the beginning of the article. This will allow the reader to find them quickly.

The work presented for review is review, it does not bring anything new to science. But such works are also valuable. Information on seed vigor assessment methods has been gathered in one place. The work is important for readers who want to get acquainted with the achievements of other researchers.

Valuable information in the work is the description of some methods used to assess the vigor of seeds and the indication of gaps in the literature on this subject. This may indicate the way for further research in the field of non-invasive quality assessment.

Introduction: Is there a research problem in the article? please formulate the research problem in the introduction.

Detailed comments:

in line: 410, 411 - to quote in the text of figure 2, it is proposed to add the designations A or B or C

 e.g. (Fig.2A)

in line: 477 - should drawing 2 be here? If so, what is it about?

In line: 439 Fig. 2. - improve the quality of drawing A, names and markings in drawings

Section 4.3. The paper discusses the method of using X-rays to evaluate seeds and the emission of odor (gases) by seeds.

Methods used for conifer seeds may also be added: the indigo carmine method.

Or maybe a cleaning method by floating in water or other liquid solutions. Separation of damaged, empty and dead seeds in water - Pravec method (Pressure or vacuum), IDS method (Incubation, Drying, Separation). These are methods that first eliminate damaged seeds.

Section 5.

I suggest that the authors of the article indicate in this part which methods (old and new) are, in their opinion, the best for assessing seed vigor and why.

References:

All listed items in the bibliography have been cited in the text.

1.    I have doubts about articles

In line: 603 - Ambrose, A., Kandpal, L. M., Kim, M. S., Lee, W.-H., & Cho, B.-K. (2016).

and

In line: 603 - Ambrose, A., Lohumi, S., Lee, W.-H., & Cho, B. K. (2016)., the same citation in the text. I propose to distinguish these articles, it is not known which article is quoted in the text. Same first author and same year. Letters a and b may be used after the year. 2.    For articles,

In line: 653 - Fan, S., Li, J., Zhang, Y., Tian, X., Wang, Q., He, X., . . . Huang, W. (2020).

In line: 659 - Fan, Y., Ma, S., & Wu, T. (2020).

same remark as in point 1. 3.    Author of publication, In line: 599 - Ahmed and In line: 789 - Raju Ahmed - this is the same person, correct in the bibliography

Ahmed, M. R., Yasmin, J., Collins, W., & Cho, B.-K. (2018). https://doi.org/https://doi.org/10.1016/j.biosystemseng.2018.09.015.

Raju Ahmed, M., Yasmin, J., Wakholi, C., Mukasa, P., & Cho, B.-K. (2020). https://doi.org/https://doi.org/10.1016/j.compag.2020.105839.

4.    Author of publication

In line: 824 - Singh Thakur P. and In line: 848 - Thakur P.S. is the same person - recall both publications correctly.

Singh Thakur, P., Tiwari, B., Kumar, A., Gedam, B., Bhatia, V., Krejcar, O., . . . Prakash, S. (2022).

Thakur, P. S., Chatterjee, A., Rajput, L. S., Rana, S., Bhatia, V., & Prakash, S. (2022).

5.    In line: 39 - should be (Apel, Hirt, 2004), not „(Apel et al., 2004).” - two person.

Author Response

Many thanks for your thoughtful and thorough review. According to your comments, we have revised the paper, please see the attachment. 

Reviewer 2 Report

This review is focused on the state of the art in relation to physiological alterations and nondestructive test methods of crops seed vigor. It achieves its objectives, and it is going to be an obligately reference in this topic.

The review is divided into three chapters: (1) Physiological changes in the decline of seed vigor caused by ROS, and how these changes affect seed vigor in turn; (2) Introduces the current vigor test methods for seeds in the process of aging or vigor decline; and (3) Some suggestions are provided for the future research on seed vigor test. Each chapter is congruent, clear and well developed; as well as link with the others.

Figure 1 and Tables provide excellent information about this topic.

The Discussion and Conclusions are congruent, clear and accurate.

The bibliographic research was exhaustive, includes the most important articles published in this topic and updated.

However, authors need to check all the spaces between the last word and the reference cited, after a point, etc. Spaces are lacking.

Author Response

Dear reviewer, many thanks for your helpful comments. We will carefully check all the spaces between the last word and the cited references, as well as the spaces after the point, and revise them according to your comments.